

# Phosphorus flow analysis of different crops in Dongying District, Shandong Province, China, 1995–2016

Huan He[1,2,*], Lvqing Zhang[3,*], Hongwei Zang[4], Mingxing Sun[5], Cheng Lv[1], Shuangshuang Li[1], Liyong Bai[1], Wenyuan Han[3] and Jiulan Dai[1]

[1] Environment Research Institute, Shandong University, QingDao, China
[2] College of Resources and Environment, Huazhong Agricultural University, WuHan, China
[3] State Key Laboratory of Agricultural Microbiology and College of Life Science and Tech-nology, Huazhong Agricultural University, WuHan, China
[4] Yantai Academy of Agricultural Sciences, YanTai, China
[5] Chinese Academy of Sciences, Institute of Geographic Sciences and Natural Resources Research, BeiJing, China
* These authors contributed equally to this work.

## ABSTRACT

Investigating the phosphorus (P) sources, pathways, and final sinks are important to reduce P pollution and improve P management. In this study, substance flow analysis (SFA) was performed for P flow analysis from 1995 to 2016 in different crops of Dongying District, a core region of the alluvial delta at the estuary of the Yellow River. The results showed that P input steadily increased from $1.48 \times 10^4$ t in 1995 to $2.16 \times 10^4$ t in 2007, and then decreased from $1.90 \times 10^4$ t in 2010 to $1.78 \times 10^4$ t in 2016. Chemical fertilizers made the highest contribution to P input. The cotton with the highest P load was on the top of P load risk ranks. More importantly, this study applied the Partial Least Squares Path Modeling (PLS-PM) model for P flow analysis and established the numerical relationship between the variables (including fertilizers, straws return-to-field, harvested grains, discarded straw, and P erosion and runoff), P use efficiency (PUE) and P load. The analysis revealed that fertilizer and crop production are the key factors affecting the PUE. Therefore, optimizing the use of P-fertilizer whilst maintaining yields can be an effective strategy to improve the local region PUE.

## INTRODUCTION

Phosphorus (P), one of the key crop nutrients, is important for crop growth and reproduction (*Smil, 2000*). Concerning material flow, P is of great significance for the stable functioning of the agricultural ecosystem (*Jiang et al., 2019a*; *Villalba et al., 2008*). However, human activities (*e.g.*, fertilizer utilization) have intensively been affecting P cycling, causing serious environmental problems, such as increasing pressure on P resources. Notably, phosphate is a non-renewable and geographically restricted resource

Corresponding author
Jiulan Dai, daijiulan@sdu.edu.cn

(*Jiang et al., 2019a*; *Simons et al., 2014*). Given that agricultural production is highly dependent on P-based fertilizers, phosphate rock has been excessively exploited. Meanwhile, excessive P utilization increases its soil accumulation, and a part of the surplus P leaches into the watery areas causing environmental issues, such as eutrophication (*Van Drecht et al., 2009*). Recently, China has become the largest producer and consumer of P fertilizer in the world, accounting for 37.5% of global production and 30% of global consumption (*Jiang et al., 2019a*; *Zhang et al., 2008*). From the perspective of P resource conservation and environmental protection, it is crucial to better manage P-based fertilizer inputs and P use efficiency (PUE).

P use efficiency (PUE), an important indicator of P usage efficiency, denotes the ratio between the P content in the harvested grains and the total P input in the agricultural systems (*Wu et al., 2016*). Improving PUE can better manage P resources and guide crops transit toward a more efficient and sustainable agricultural model (*Zheng et al., 2017*). Meanwhile, studies based on PUE can guide policymakers and local farmers to develop eco-agriculture and green agriculture.

To estimate PUE in agricultural soil, it is essential to quantify and trace P cycling in agricultural systems (*Wu et al., 2016*). There are several quantitative methods for calculating the P amount based on Geographic Information System (GIS) (*Eastman et al., 2010*) and P loss formulas (*Leone et al., 2008*). However, those methods only consider the P amount but not the P flows. Therefore, these calculations fail to trace P flows and interpret their relationship with human activities. The mass-balance methods, which can analyze and quantify the P flows, are appropriate for PUE estimation. In particular, substance flow analysis (SFA) systematically connects the P sources, pathways, and final sinks, and therefore can efficiently quantify the P balance within the agricultural system (*Wang et al., 2020*). Substance flow analysis (SFA) has been widely used for PUE estimation and P flow analysis to help policymakers or local farmers to develop the proper P management system. For example, *Yuan et al. (2011)* used SFA to establish an anthropogenic phosphorus flows model within a socioeconomic system in Chaohu City over 2008 and found that fertilizers utilization in agricultural soil was the most important source of P load on local surface water. Eventually, they suggested limiting fertilizer use for ecological agriculture to reduce the eutrophication of water bodies. In New Zealand, *Li et al. (2017)* used SFA to estimate P recovery from pollution sources and suggested measures to improve PUE.

The Yellow River Delta (YRD), one of the fastest-growing deltas in the world, covers an area of 5,400 km$^2$, of which 5,200 km$^2$ area is located in the Dongying District, which is a major agricultural base in Shandong Province, China (*He et al., 2020b*). The soil in the Dongying District has high salinity and low nutrients (*Guo et al., 2020*); therefore, excess P fertilizer is applied to achieve optimal crop yields, causing P pollution and low PUE. Additionally, the crop-planting pattern in the Dongying District has changed dramatically in the past 20 years from the traditional crop-planting pattern of "grain crops-economic crops" to "grain crops-economic crops-feed crops-energy crops" (*Gu & Wang, 2008*). In general, the cultivation of feed and energy crops demands more fertilizer than grain and economic crops (*Huan et al., 2020*). Thus, the current crop-planting pattern in the

Dongying District forces excessive P fertilizer usage, increasing the risk of P pollution in agricultural soils. Therefore, it is urgent to analyze the P inputs, outputs, losses, and accumulations in different crops of the Dongying District and find the strategies to improve the current situation.

Accordingly, our study structured the P flow frame in the regional agricultural system by systematic evaluation of P flows, PUE, and P load risk rank. By SFA approaches (*Wu et al., 2016*), we quantified the P inputs, outputs, losses, and accumulations in seven crops (wheat, maize, rice, soybean, peanut, cotton, and fruit-vegetable) from 1995 to 2016 (over 22 years) in the Dongying District. Our results can help manage the sustainable agricultural development in the Dongying District.

## MATERIALS AND METHODS

### Study area

Dongying District, located in the northeast of Shandong Province, is a core region of the alluvial delta at the estuary of the Yellow River. The region has a warm temperate continental monsoon climate with an average temperature of 14.3 °C and annual rainfall of 684.7 mm (*Boss et al., 2011*). Presently, the total area of the city is ~8,243.26 km$^2$ including 51.72% as agricultural land (*China Statistics Press (2018)*). The agricultural soils mainly contain tidal and saline-alkali soils (*Ottinger et al., 2013*). The soil of the region has high salinity and low nutrients (*Guo et al., 2020*). Additionally, the planting structure in the Dongying District had significantly changed greatly between 1995 and 2016 (Fig. 1). The traditional crops of the region include the grain (*e.g.*, wheat and maize) and economic (*e.g.*, cotton) crops. Since the year 2000, several newly formed lands have been developed and utilized as agricultural soils and the farmers are encouraged to cultivate economic crops (*e.g.*, cotton) (*Gu & Wang, 2008*). The crops analyzed in our study, including wheat, maize, rice, soybean, peanut, cotton and fruit-vegetable, are under conventional farming, which mainly relies on chemical intervention to provide crop nutrition.

### System definition

We used SFA following a previous work of *Wu et al. (2016)* with minor modifications to analyze the P flows in seven crops of Dongying District, including wheat, maize, rice, soybean, peanut, cotton, and fruit-vegetable. The frame of P flows is shown in Fig. 2. The studied period covered from 1995 to 2016. For the simplification of statistical results, the years 1995, 1998, 2001, 2004, 2007, 2010, 2013, 2016 were chosen to represent change every three years.

In our investigation, we assumed that the agricultural ecosystem in the Dongying District was in a steady state. Concerning the available data, atmospheric deposition, seeds, chemical fertilizers, pesticides, straw return-to-field, irrigation, resident and livestock excrement were considered as P inputs to the crops. In addition, harvested grains, discarded straw, erosion and runoff were recognized as P outputs. The difference between P inputs and outputs was considered as soil P accumulation.

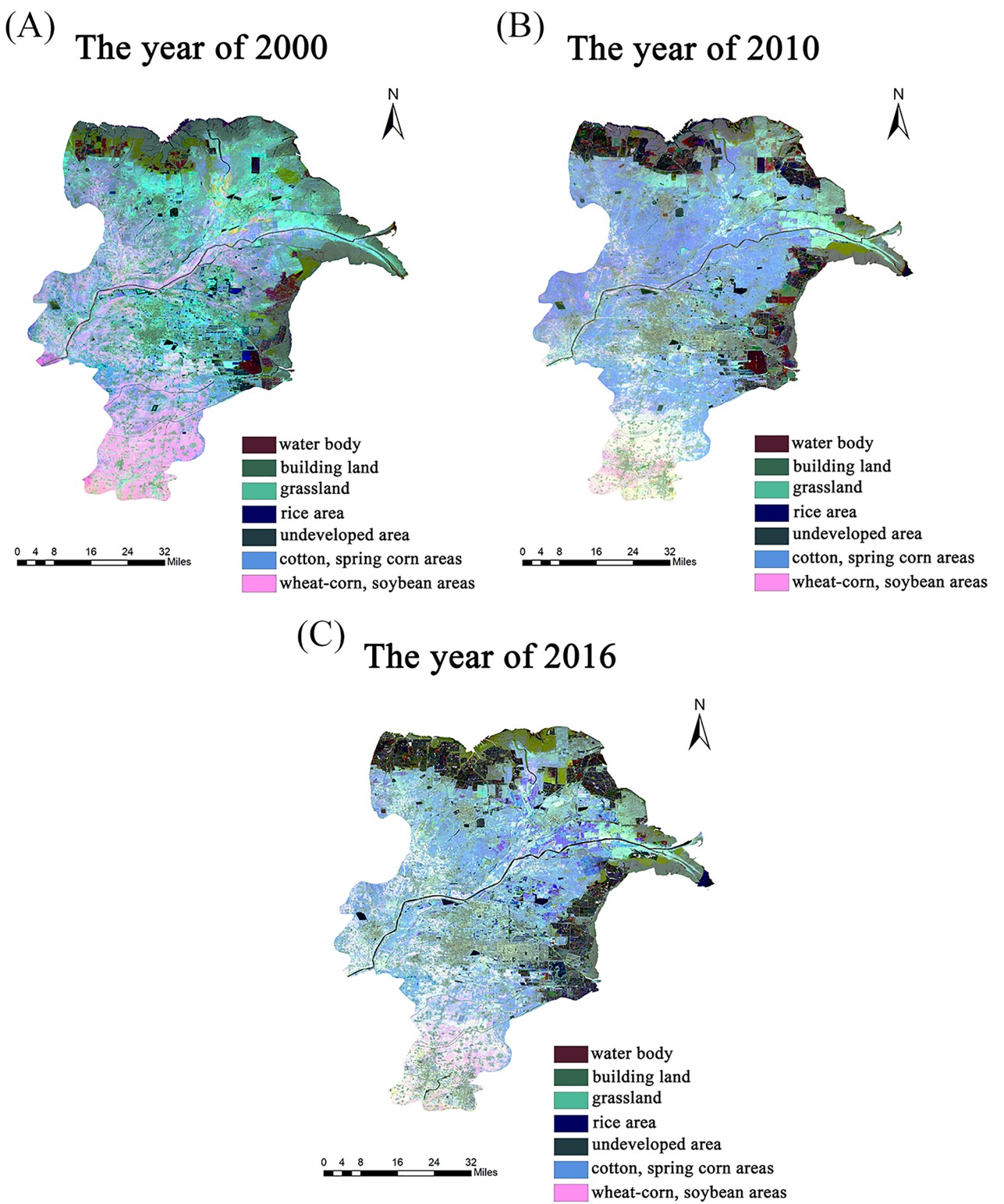

**Figure 1 Land uses and cropping covers in Dongying district.** (A) Land uses and cropping covers in Dongying district for the years of 2000; (B) land uses and cropping covers in Dongying district for the years of 2010; (C) land uses and cropping covers in Dongying district for the years of 2016.

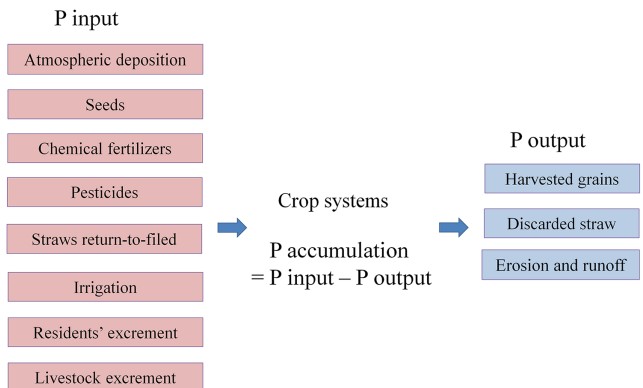

**Figure 2 The diagram of phosphorus (P) flows in crops of Dongying district.** The figure showed the frame of P flows in crops of Dongying district. Concerning the available data, atmospheric deposition, seeds, chemical fertilizers, pesticides, straw return-to-field, irrigation, resident and livestock excrement were considered as P inputs to the cropping systems. In addition, harvested grains, harvested straw, erosion and runoff were recognized as P outputs. The difference between P inputs and outputs was considered as soil P accumulation.

## Quantification

Based on the mass balance principle, the equation "P accumulation = P input – P output" was used to describe P flows in the above crops. Additionally, a dynamic model was developed to quantify annual P flows and distinguish the changes during 22 years. The detailed model equations of the P inputs and outputs of the system are described in Table 1.

In this study, we quantified the PUE, P nutrient load, and the risk index of P load in the seven crops of the Dongying District from 1995 to 2016. Specifically, PUE described as the ratio between the P content from the harvested grain and the total P inputs (*Senthilkumar et al., 2012*; *Wu et al., 2016*), was used to assess the P use intensity and efficiency in the agricultural soil systems. The P nutrient load and the risk index of P load can evaluate the healthy status of soil in the agricultural systems (Table 1). Here, P load refers to the difference between P inputs and outputs per area (*Dambeniece-Migliniece, Veinbergs & Lagzdins, 2018*). The risk index of P load was estimated according to the ratio between P content of fertilizers (chemical fertilizer and excrement) used and the appropriate amount of P fertilizers used in local farmland (Table 1, Eq. (14)) (*Oenema et al., 2004*).

## Structural equation model

Partial least squares path modeling (PLS-PM) is a variance-based structural equation modeling technique that relies on an alternating least squares algorithm (*Henseler, 2018*). It is widely used in various disciplines as an effective tool to analyze the relationships and influence of different aspects on an observed phenomenon. Recently, the improved PLS-PM method has been used for the analysis of cause-effect relationships in confirmatory and explanatory research (*Benitez et al., 2020*). In this study, factors with higher P input proportions, *i.e.*, chemical fertilizers, resident and livestock excrement, and straw return-to-field, were selected as block variables. Furthermore, the factors of P

**Table 1  Equations of the P flow calculation of the crops.**

| P flows | Calculation formula | |
|---|---|---|
| **P input** | | |
| $P^{atmos}$ | $P_i^{atmos} = A_i^{area} w^{wind} + A_i^{area} A^{preci} r^{rain}$ | Eq. (1) |
| $P^{seed}$ | $P_i^{seed} = A_i^{area} w_i^{seed} r_i^{seed}$ | Eq. (2) |
| $P^{chem}$ | $P_i^{chem} = B_i^{com} r^{com} + B_i^{phos} r^{phos}$ | Eq. (3) |
| $P^{pest}$ | $P_i^{pest} = B_i^{pest} r^{pest}$ | Eq. (4) |
| $P^{retured\ straw}$ | $P_i^{retured\ straw} = B_i^{grain} r_i^{grain-straw} r_i^{straw} r^{retured\ straw}$ | Eq. (5) |
| $P^{resid}$ | $P_i^{resid} = B^{resid} w^{resid} r^{resid} \frac{A_i^{area}}{A^{area}}$ | Eq. (6) |
| $P^{livestock}$ | $P_i^{livestock} = B^{livestock} w^{livestock} r^{livestock} \frac{A_i^{area}}{A^{area}}$ | Eq. (7) |
| $P^{irrig}$ | $P_i^{irrig} = A_i^{area} w^{irrig}$ | Eq. (8) |
| **P output** | | |
| $P^{grain}$ | $P_i^{grain} = B_i^{grain} r_i^{grain}$ | Eq. (9) |
| $P^{discarded\ straw}$ | $P_i^{discarded\ straw} = B_i^{grain} r_i^{grain-straw} r_i^{straw} \left( 1 - r^{retured\ straw} \right)$ | Eq. (10) |
| $P^{loss}$ | $P_i^{loss} = A_i^{area} r_i^{loss}$ | Eq. (11) |
| **PUE** | $PUE = P_i^{grain} / P_i^{input}$ | Eq. (12) |
| **P load** | $P\ load = (P^{input} - P^{output})/A^{area}$ | Eq. (13) |
| **The risk model of P load (r)** | $r = (P^{Chem}/Fp + (P^{resid} + P^{livestock})/Lp)/2$ | Eq. (14) |

Notes:
Here,
(1) For Eq. (1)

$$P_i^{atmos} = A_i^{area} w^{wind} + A_i^{area} A^{preci} r^{rain} \tag{1}$$

Here,
$P_i^{atmos}$: Atmospheric deposition P of wheat, maize, rice, soybean, peanut, cotton, fruit-vegetable, respectively (t);
$A_i^{area}$: Sown areas of wheat, maize, rice, soybean, peanut, cotton, and fruit-vegetable, respectively (ha);
$w^{wind}$: Average wind erosion intensity per sown area (kg/ha);
$A^{preci}$: Precipitaiton (mm/ha);
$r^{rain}$: P content in the rail fall (mg/L);
(2) For Eq. (2)

$$P_i^{seed} = A_i^{area} w_i^{seed} r_i^{seed} \tag{2}$$

Here,
$P_i^{seed}$: P content of the seeds of wheat, maize, rice, soybean, peanut, cotton, fruit-vegetable, respectively (t);
$A_i^{area}$: Sown areas of wheat, maize, rice, soybean, peanut, cotton, fruit-vegetable, respectively (ha);
$w_i^{seed}$: Seed amount per sown area of wheat, maize, rice, soybean, peanut, cotton, fruit-vegetable, respectively (kg/m$^2$);
$r_i^{seed}$: P-containing rates of seeds of wheat, maize, rice, soybean, peanut, cotton, fruit-vegetable, respectively (%);
(3) For Eq. (3)

$$P_i^{chem} = B_i^{com} r^{com} + B_i^{phos} r^{phos} \tag{3}$$

$P_i^{chem}$: P content of chemical fertilizers used in wheat, maize, rice, soybean, peanut, cotton, fruit-vegetable, respectively (t);
$B_i^{com}$: Amount of compound fertilizer used in wheat, maize, rice, soybean, peanut, cotton, fruit-vegetable, respectively (t);
$r^{com}$: P-containing rate of the compound fertilizer (%);
$B_i^{phos}$: Amount of phosphate fertilizer used in wheat, maize, rice, soybean, peanut, cotton, fruit-vegetable, respectively (t);
$r^{phos}$: P-containing rate of the phosphate fertilizer (%);
(4) For Eq. (4)

$$P_i^{pest} = B_i^{pest} r^{pest} \tag{4}$$

$P_i^{pest}$: P content of the pesticide used in wheat, maize, rice, soybean, peanut, cotton, fruit-vegetable, respectively (t);
$B_i^{pest}$: Amount of pesticide used in wheat, maize, rice, soybean, peanut, cotton, fruit-vegetable, respectively (t);
$r^{pest}$: P-containing rate of the pesticide (%);
(5) For Eq. (5)

$$P_i^{retured\ straw} = B_i^{grain} r_i^{grain-straw} r_i^{straw} r^{retured\ straw} \tag{5}$$

$P_i^{retured\ straw}$: P content of the straw return-to-field of wheat, maize, rice, soybean, peanut, cotton, fruit-vegetable, respectively (t);
$B_i^{grain}$: Harvest of wheat, maize, rice, soybean, peanut, cotton, fruit-vegetable, respectively (t);
$r_i^{grain-straw}$: Grain to straw ratio of wheat, maize, rice, soybean, peanut, cotton, fruit-vegetable, respectively;
$r_i^{straw}$: P-containing rate of straws of wheat, maize, rice, soybean, peanut, cotton, fruit-vegetable, respectively (%);
$r^{retured\ straw}$: straw return-to-field ratio (%);
(6) For Eq. (6)

$$P_i^{resid} = B^{resid} w^{resid} r^{resid} \frac{A_i^{area}}{A^{area}} \tag{6}$$

$P_i^{resid}$: P content of the residents' excrement applied to the field of wheat, maize, rice, soybean, peanut, cotton, fruit-vegetable, respectively (t);
$B^{resid}$: The population of residents;
$w^{resid}$: P content of residents' excrement (kg);
$r^{resid}$: Proportion of residents' excrement applied to the field (%);
$A_i^{area}$: Sown areas of wheat, maize, rice, soybean, peanut, cotton, and fruit-vegetable, respectively (ha);
$A^{area}$: Sown area of all crop (ha);
(7) For Eq. (7)

$$P_i^{livestock} = B^{livestock} w^{livestock} r^{livestock} \frac{A_i^{area}}{A^{area}} \tag{7}$$

$P_i^{livestock}$: P content of the livestock excrement applied to the field of wheat, maize, rice, soybean, peanut, cotton, fruit-vegetable, respectively (t);
$B^{livestock}$: The amount of pig, cattle and sheep, respectively;
$w^{livestock}$: P content of livestock excrement (kg);
$r^{livestock}$: Proportion of livestock excrement applied to the field (%);
$A_i^{area}$: Sown areas of wheat, maize, rice, soybean, peanut, cotton, and fruit-vegetable, respectively (ha);
$A^{area}$: Sown area of all crop (ha);
(8) For Eq. (8)

$$P_i^{irrig} = A_i^{area} w^{irrig} \tag{8}$$

$P_i^{irrig}$: P content of irrigation in wheat, maize, rice, soybean, peanut, cotton, and fruit-vegetable, respectively (t);
$A_i^{area}$: Sown areas of wheat, maize, rice, soybean, peanut, cotton, and fruit-vegetable, respectively (ha);
$w^{irrig}$: P content of irrigation per sown area of wheat, maize, rice, soybean, peanut, cotton, fruit-vegetable, respectively (kg/ha);
(9) For Eq. (9)

$$P_i^{grain} = B_i^{grain} r_i^{grain} \tag{9}$$

$P_i^{grain}$: P content of the harvested grains of wheat, maize, rice, soybean, peanut, cotton, and fruit-vegetable, respectively (t);
$B_i^{grain}$: Harvest of wheat, maize, rice, soybean, peanut, cotton, fruit-vegetable, respectively (t);
$r_i^{grain}$: P-containing rate of grains of wheat, maize, rice, soybean, peanut, cotton, fruit-vegetable, respectively (%);
(10) For Eq. (10)

$$P_i^{discarded\ straw} = B_i^{grain} r_i^{grain-straw} r_i^{straw} \left(1 - r^{retured\ straw}\right) \tag{10}$$

$P_i^{discarded\ straw}$: P content of the discarded straw of wheat, maize, rice, soybean, peanut, cotton, fruit-vegetable, respectively (t);
$B_i^{grain}$: Harvest of wheat, maize, rice, soybean, peanut, cotton, fruit-vegetable, respectively (t);
$r_i^{grain-straw}$: Grain to straw ratio of wheat, maize, rice, soybean, peanut, cotton, fruit-vegetable, respectively;
$r_i^{straw}$: P-containing rate of straws of wheat, maize, rice, soybean, peanut, cotton, fruit-vegetable, respectively (%);
$r^{retured\ straw}$: Straw return-to-field ratio (%);
(11) For Eq. (11)

$$P_i^{loss} = A_i^{area} r_i^{loss} \tag{11}$$

$P_i^{loss}$: P content from erosion and runoff in the field of wheat, maize, rice, soybean, peanut, cotton, fruit-vegetable, respectively (t);
$A_i^{area}$: Sown areas of wheat, maize, rice, soybean, peanut, cotton, and fruit-vegetable, respectively (ha);
$r_i^{loss}$: P loss content per unit of the area of wheat, maize, rice, soybean, peanut, cotton, fruit-vegetable, respectively (kg/ha).
(12) For Eq. (12)

$$PUE = P_i^{grain} / P_i^{input} \tag{12}$$

The PUE was described as the ratio between the P content from the harvested grain and the total P inputs in studied system.
$P_i^{grain}$: P content of the harvested grains of wheat, maize, rice, soybean, peanut, cotton, and fruit-vegetable, respectively (t);
$P_i^{input}$: P input of wheat, maize, rice, soybean, peanut, cotton, and fruit-vegetable, respectively (t).
(13) For Eq. (13)

$$P\ load = P_i^{input} - P_i^{output} / A_i^{area} \tag{13}$$

P load is used to assess the health of the soil system, and it refers to the difference between P inputs and P outputs per area.
$P_i^{input}$: P input of wheat, maize, rice, soybean, peanut, cotton, and fruit-vegetable, respectively (t);
$P_i^{output}$: P output of wheat, maize, rice, soybean, peanut, cotton, and fruit-vegetable, respectively (t).
$A_i^{area}$: Sown areas of wheat, maize, rice, soybean, peanut, cotton, and fruit-vegetable, respectively (ha);
(14) For Eq. (14)

$$R_p = \left( P_i^{chem} / F_p + (P_i^{resid} + P_i^{livestock}) / L_p \right) / 2 \tag{14}$$

$P_i^{chem}$: P content of chemical fertilizers used in wheat, maize, rice, soybean, peanut, cotton, fruit-vegetable, respectively (t);
$P_i^{livestock}$: P content of the livestock excrement applied to field of wheat, maize, rice, soybean, peanut, cotton, fruit-vegetable, respectively (t);
$P_i^{resid}$: P content of the residents' excrement applied to field of wheat, maize, rice, soybean, peanut, cotton, fruit-vegetable, respectively (t);
$F_p$ is the appropriate amount of P fertilizers (70 kg/ha);
$L_p$ is the maximum P content in excrement used (35 kg/ha).

output including harvested grains, discarded straw and P erosion and runoff were considered as block variables. Thus, the block variables of the PLS-PM in our study included "fertilizers (residents' excrement, livestock excrement, and chemical fertilizers)", "straws return-to-field", "harvested grains", "discarded straw" and "P erosion and runoff".

In the aspect of accuracy of PLS-PM, the loadings of those block variables were selected with the threshold 0.7, because the indicators with loadings >0.7 were regarded as adequate indicators for the corresponding block variables (*Wang et al., 2016*). In our study, the loadings of the five block variables were >0.7. Moreover, the PLS-PM model was performed in the plspm package (version 0.4.9) in R (version 4.0.3) and was validated by 1,000 bootstraps. The relationships between the five block variables, PUE and P loads were described by path coefficients, which was considered significant when $P < 0.05$.

## Data collection

To quantify the P flows in agricultural soil systems of the Dongying District, the data was collected from the official statistical database, interviews, questionnaires, and published data. The statistical Yearbooks of Dongying District (*China Statistics Press, 1995*, *1998*, *2001*, *2004*, *2007*, *2010*, *2013*, *2016*) were used to obtain the precipitation, crop areas, harvested grains, chemical fertilizers, pesticides used in the field, and the amount and area of straw returning to the field. These data are listed in Table 1 and the File S1. The Table S1 provides the details about data acquisition.

In addition, parameters such as the average wind erosion intensity ($w^{wind}$; 0.05 Kg/Ha; *Smil, 2000*), P content in the rail fall ($r^{rain}$; 0.01 mg/L; *Smil, 2000*), P-containing rate of the compound fertilizer ($r^{com}$; 11.71%; *Wu et al., 2016*), P-containing rate of the phosphate fertilizer ($r^{phos}$; 43.66%; *Wu et al., 2016*), P-containing rate of the pesticide ($r^{pest}$; 3.30%; *Yan et al., 2008*), P content of residents and livestock excrement ($w^{resid} = w^{livestock}$; 0.73 kg; *Wu et al., 2016*), the proportion of the excrement applied to the field ($r^{resid} = r^{livestock}$; 95%; *Wu et al., 2016*), and P content of irrigation per sown area of crop ($w^{irrig}$; 0.45 kg ha$^{-1}$; *Chen, Chen & Sun, 2008*) were determined. The P-containing rates of wheat, maize, rice, soybean, peanut, cotton, and fruit-vegetable seeds or grains ($r_i^{seed} = r_i^{grain}$) were assumed to be 0.5%, 0.4%, 0.4%, 0.6%, 0.5%, 0.78%, and 0.13% (*Wu et al., 2016*), respectively. The P-containing rates of straw ($r_i^{straw}$) of wheat, maize, rice, soybean, peanut, cotton, fruit-vegetable were 0.08%, 0.15%, 0.13%, 0.20%, 0.16%, 0.15%, and 0%, respectively (*Smil, 2000*; *Wu et al., 2016*). The grain to straw ratios ($r_i^{grain-straw}$) of wheat, maize, rice, soybean, peanut, cotton, and fruit-vegetable were 1.10, 3.00, 1.00, 1.60, 1.50, 3.00, and 0.00, respectively (*Smil, 2000*; *Wu et al., 2016*). From 1995 to 1998, straw return-to-field ratios ($r^{retured\ straw}$) of wheat, maize, rice, peanut and cotton were 0.5, which later increased to 0.85 from 2001 onwards. Comparatively, the straw return-to-field ratio ($r^{retured\ straw}$) of soybean was 0.16, 0.16, 0.56, 0.56, 0.56, 0.56, 0.56, and 0.56 in years of 1995, 1998, 2001, 2004, 2007, 2010, 2013, and 2016, respectively (File S1). These data and respective sources are listed in Table S1.

Furthermore, the P loss data was variable from 1995 to 2016. Notably, the P loss is associated with P erosion and runoff and is tough to determine (*Jiang et al., 2019b*; *Schroder et al., 2011*). Therefore, this study used the P loss rates based on the published

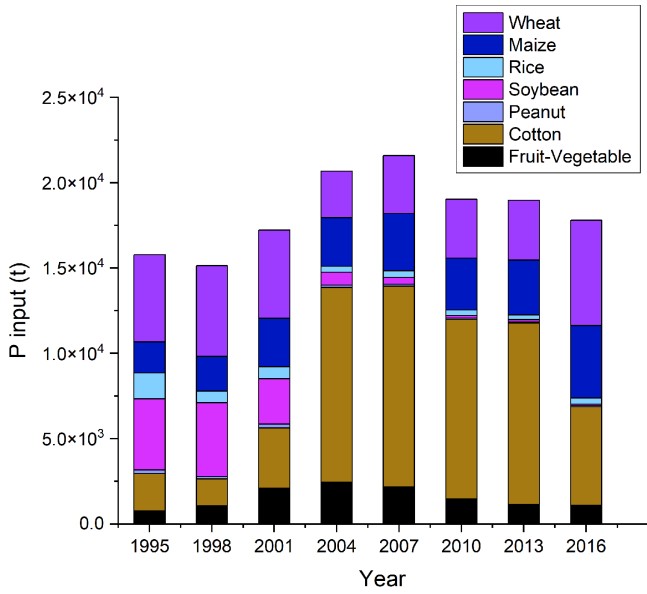

**Figure 3 Phosphorus (P) input into the seven crops in Dongying district from 1995 to 2016.** The crops include wheat, maize, rice, soybean, peanut, cotton and fruit-vegetable.

literature and inferred that it changed with the applied fertilizers (*Chen, Chen & Sun, 2008*; *Fan et al., 2009*; *Senthilkumar et al., 2012*; *Wu et al., 2016*; *Yan et al., 1999*). The values related with P loss were shown in File S1.

## Statistical analysis

In this study, the parameters of P input, P output, PUE, P nutrient load, and the risk index of P load in the crops of the Dongying District were estimated. The model equations of these parameters mentioned above are described in Table 1 and File S1. In addition, the model PLS-PM was performed in plspm package in R (version 4.0.3). The block variables from P inputs were selected based on their P input proportions. Thus, the factors with higher P input proportions, *i.e.*, chemical fertilizers, resident and livestock excrement, and straw return-to-field, were selected as block variables. Furthermore, the correlation between P input from fertilization and PUE of different crops was studied by "cor" function (method = "pearson") in stats package in R (version 4.0.3).

## RESULTS

### P inputs

Atmospheric deposition, seeds, chemical fertilizers, pesticides, straw return-to-filed, resident and livestock excrement, and irrigation were considered as P inputs to study the different crops. Of the Dongying District, P input of agricultural soil systems increased steadily from $1.48 \times 10^4$ t in 1995 to $2.16 \times 10^4$ t in 2007, while it decreased from $1.90 \times 10^4$ t in 2010 to $1.78 \times 10^4$ t in 2016 (Fig. 3). Specifically, compared with other crops, cotton had the highest P input from 2004 to 2013 (Fig. 3). It rose by 5.36 folds from $2.19 \times 10^3$ t in 1995 to $1.17 \times 10^4$ t in 2007 and maintained high levels until 2013.

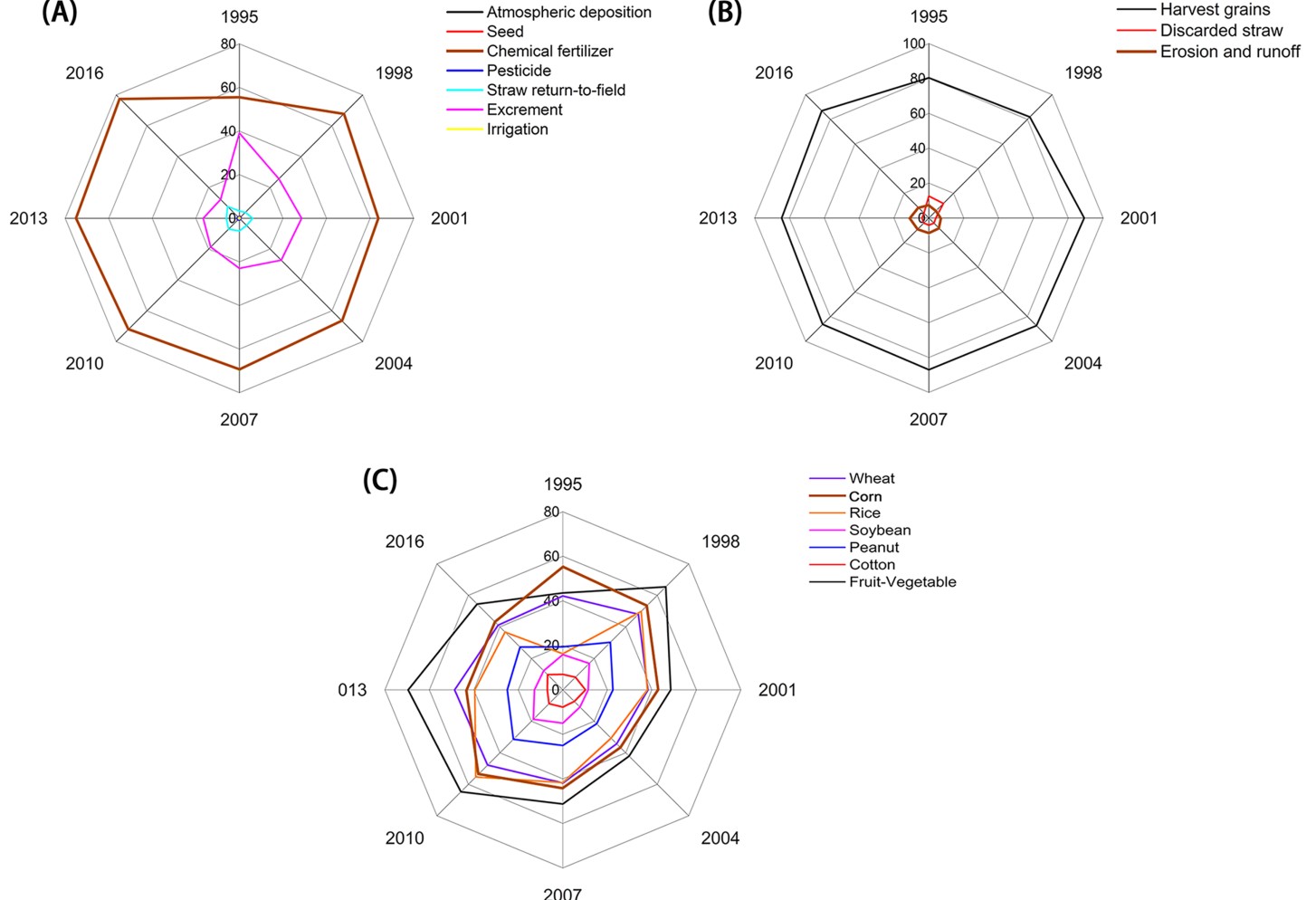

**Figure 4** (A) **The proportion of the phosphorus (P) input factors (A), output factors (B) and the phosphorus use efficiency (PUE) (C) in the different crops in Dongying district from 1995 to 2016.** The P input factors include atmospheric deposition, seeds, chemical fertilizers, pesticides, straw return-to-filed, resident and livestock excrement, and irrigation. The P output factors include harvested grains, harvested straw, and erosion and runoff. The seven crops include wheat, maize, rice, soybean, peanut, cotton and fruit-vegetable.

Furthermore, the P input of the soybean decreased sharply from $4.33 \times 10^3$ t in 1998 to 65 t in 2016, which was caused by decreasing soybean cultivation. Only a few farmers were willing to plant soybean in the Dongying District, due to tedious harvesting, storage and post-harvest management of and low economic efficiency (*Bern, Hanna & Wilcke, 2008*). Since then, the gap between supply and demand of soybean has been expanding, while the supply of maize exceeded demand in the Dongying District. We recognized that the crop structure of the Dongying District is unbalanced making it unfavorable for sustained agricultural development.

Concerning the source of P inputs, the chemical fertilizers contributed the highest from 1995 to 2016, followed by the excrement and straw return-to-field (Fig. 4A). Moreover, the proportion of P input from chemical fertilizers increased from 55.4% in

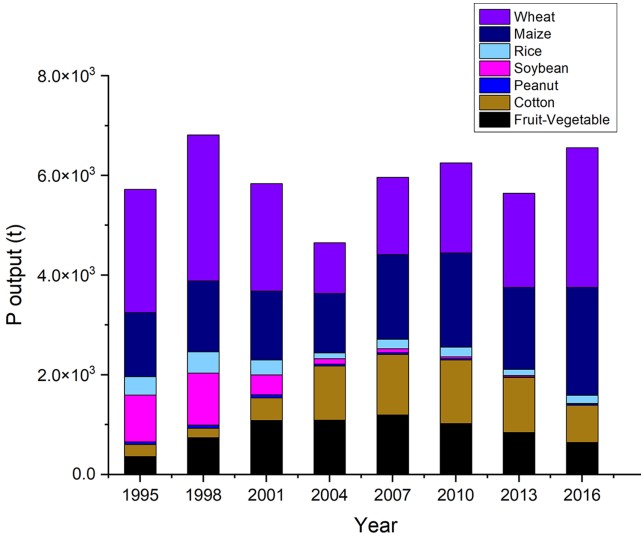

**Figure 5 Phosphorus (P) output from the seven crops in Dongying district from 1995 to 2016.** The crops include wheat, maize, rice, soybean, peanut, cotton and fruit-vegetable.

1995 to 78.1% in 2016, while the excrement proportion decreased from 39.1% to 12.2% in the same period (Fig. 4A).

## P outputs

Harvested grains, discarded straw and erosion and runoff were considered as P outputs. Phosphorus (P) amount from harvested grains accounted for 80–90% of the total P output (Fig. 4B). It decreased from $5.59 \times 10^3$ t in 1998 to $4.06 \times 10^3$ t in 2004 and then increased to $5.70 \times 10^3$ t in 2016 (Fig. 5). The trend of change in total P output is consistent with that of harvested grains, which decreased from $6.81 \times 10^3$ t in 1998 to $4.65 \times 10^3$ t in 2004, and then increased to $6.56 \times 10^3$ t in 2016 (Table 2).

Compared with other crops, wheat and maize had the highest P output during the 22 years (Fig. 5). Specifically, the total P output of the wheat had the highest proportion 21–43%, followed by the maize 20–33%. The P output of the wheat first dropped from $2.47 \times 10^3$ t in 1995 to $1.02 \times 10^3$ t in 2004, and then gradually increased from $1.55 \times 10^3$ t in 2007 to $2.81 \times 10^3$ t in 2016. The P output of the maize showed a similar trend first decreasing from $1.29 \times 10^3$ t in 1995 to $1.19 \times 10^3$ t in 2004 and then increasing from $1.70 \times 10^3$ t in 2007 to $2.16 \times 10^3$ t in 2016.

## P load and risk index

In this study, we used the P load index to indicate the soil health status. We found that the cotton had the highest P load value ranging from 63.42 kg/ha (in 1998) to 104.62 kg/ha (in 2004), followed by the soybean with a P load value of 49.47 kg/ha (in 1998)–87.23 kg/ha (in 2004) (Fig. 6). Next, we estimated the P load risk index of these crops in the study areas (Table 3), and found that it followed the trend of cotton > soybean > wheat > Fruit-vegetable > maize, rice, and peanut. The cotton was on the top of the P load risk ranks, in the II and III classes during the 22 years study period.

**Table 2 Phosphorus input and output in different crop s of Dongying district from 1995 to 2016.**

| P Input (t) | | Wheat | Maize | Rice | Soybean | Peanut | Cotton | Fruit-Vegetable |
|---|---|---|---|---|---|---|---|---|
| Atmospheric deposition | | | | | | | | |
| 1995 | | 9.48 | 4.68 | 0.98 | 6.28 | 0.41 | 3.03 | 1.66 |
| 1998 | | 9.71 | 4.35 | 1.43 | 6.39 | 0.31 | 2.07 | 2.39 |
| 2001 | | 6.86 | 4.24 | 1.04 | 2.93 | 0.35 | 3.41 | 3.48 |
| 2004 | | 3.58 | 4.41 | 0.51 | 0.87 | 0.20 | 11.25 | 4.00 |
| 2007 | | 3.77 | 4.40 | 0.51 | 0.38 | 0.16 | 9.76 | 3.12 |
| 2010 | | 5.64 | 6.18 | 0.69 | 0.24 | 0.16 | 13.27 | 3.29 |
| 2013 | | 4.28 | 5.54 | 0.42 | 0.17 | 0.08 | 10.50 | 2.10 |
| 2016 | | 9.35 | 8.87 | 0.69 | 0.09 | 0.15 | 6.45 | 2.46 |
| Seeds | | | | | | | | |
| 1995 | | 82.56 | 5.43 | 1.90 | 43.76 | 3.33 | 2.28 | 1.25 |
| 1998 | | 90.87 | 5.43 | 2.97 | 47.79 | 2.70 | 1.68 | 1.94 |
| 2001 | | 70.97 | 5.85 | 2.38 | 24.23 | 3.43 | 3.06 | 3.12 |
| 2004 | | 28.29 | 4.65 | 0.90 | 5.47 | 1.49 | 7.70 | 2.74 |
| 2007 | | 39.45 | 6.14 | 1.19 | 3.15 | 1.57 | 8.85 | 2.83 |
| 2010 | | 47.64 | 6.97 | 1.30 | 1.65 | 1.29 | 9.72 | 2.41 |
| 2013 | | 46.98 | 8.12 | 1.03 | 1.46 | 0.87 | 9.99 | 2.00 |
| 2016 | | 73.57 | 9.31 | 1.21 | 0.55 | 1.09 | 4.40 | 1.68 |
| Chemical fertilizers | | | | | | | | |
| 1995 | | 2806.18 | 1011.97 | 239.08 | 2718.14 | 98.36 | 1470.88 | 403.75 |
| 1998 | | 3522.47 | 1112.02 | 416.13 | 3262.43 | 89.08 | 1207.57 | 697.12 |
| 2001 | | 3263.25 | 1384.45 | 389.94 | 1911.21 | 132.16 | 2564.45 | 1308.83 |
| 2004 | | 1728.73 | 1398.54 | 191.22 | 548.51 | 74.56 | 8374.18 | 1487.07 |
| 2007 | | 2267.25 | 1728.70 | 237.07 | 295.71 | 73.63 | 9019.87 | 1444.02 |
| 2010 | | 2421.11 | 1582.52 | 216.52 | 125.29 | 50.56 | 8328.13 | 1033.46 |
| 2013 | | 2475.41 | 1857.49 | 176.02 | 111.43 | 34.93 | 8739.33 | 873.91 |
| 2016 | | 4920.52 | 2744.04 | 263.67 | 53.54 | 55.73 | 4926.71 | 938.26 |
| Pesticides | | | | | | | | |
| 1995 | | 38.46 | 18.97 | 3.99 | 25.48 | 1.64 | 12.28 | 6.74 |
| 1998 | | 44.88 | 20.11 | 6.60 | 29.50 | 1.41 | 9.58 | 11.06 |
| 2001 | | 33.60 | 20.78 | 5.08 | 14.34 | 1.72 | 16.70 | 17.04 |
| 2004 | | 21.37 | 26.32 | 3.07 | 5.16 | 1.20 | 67.12 | 23.84 |
| 2007 | | 26.60 | 31.06 | 3.62 | 2.66 | 1.12 | 68.83 | 22.04 |
| 2010 | | 39.29 | 43.09 | 4.81 | 1.71 | 1.12 | 92.49 | 22.95 |
| 2013 | | 41.15 | 53.33 | 4.07 | 1.60 | 0.81 | 100.98 | 20.19 |
| 2016 | | 54.17 | 51.42 | 4.00 | 0.50 | 0.85 | 37.40 | 14.25 |
| Straw return-to-field | | | | | | | | |
| 1995 | | 176.38 | 226.65 | 37.24 | 34.39 | 8.70 | 41.04 | 0.00 |
| 1998 | | 208.71 | 246.04 | 51.98 | 37.99 | 10.04 | 33.61 | 0.00 |
| 2001 | | 275.70 | 465.75 | 68.07 | 55.29 | 20.18 | 162.91 | 0.00 |
| 2004 | | 130.00 | 400.12 | 26.82 | 15.37 | 10.97 | 381.71 | 0.00 |

| Table 2 (continued) | | | | | | | |
|---|---|---|---|---|---|---|---|
| **P Input (t)** | | **Wheat** | **Maize** | **Rice** | **Soybean** | **Peanut** | **Cotton** | **Fruit-Vegetable** |
| | 2007 | 198.00 | 569.64 | 43.11 | 10.89 | 11.82 | 411.64 | 0.00 |
| | 2010 | 229.00 | 628.06 | 45.55 | 5.391 | 8.95 | 420.11 | 0.00 |
| | 2013 | 237.00 | 531.09 | 27.08 | 3.27 | 5.04 | 339.45 | 0.00 |
| | 2016 | 353.00 | 702.86 | 34.62 | 1.46 | 7.81 | 255.05 | 0.00 |
| | Residents and livestock excrement | | | | | | | |
| | 1995 | 1976.20 | 533.32 | 223.43 | 1326.07 | 87.96 | 653.63 | 356.65 |
| | 1998 | 1412.55 | 633.06 | 207.83 | 928.64 | 44.49 | 301.55 | 348.16 |
| | 2001 | 1511.62 | 934.79 | 228.44 | 645.23 | 77.43 | 751.18 | 766.77 |
| | 2004 | 810.83 | 998.64 | 116.29 | 195.83 | 45.34 | 2546.40 | 904.37 |
| | 2007 | 845.77 | 987.71 | 115.06 | 84.48 | 35.74 | 2188.90 | 700.86 |
| | 2010 | 687.78 | 754.30 | 49.88 | 23.04 | 12.64 | 1618.96 | 401.80 |
| | 2013 | 684.30 | 740.25 | 56.47 | 22.18 | 11.21 | 1401.71 | 236.37 |
| | 2016 | 737.32 | 699.98 | 61.16 | 9.32 | 9.68 | 509.17 | 145.89 |
| | Irrigation | | | | | | | |
| | 1995 | 23.28 | 11.48 | 2.42 | 15.42 | 0.99 | 7.43 | 4.08 |
| | 1998 | 23.90 | 10.71 | 3.52 | 15.71 | 0.75 | 5.10 | 5.89 |
| | 2001 | 19.63 | 12.14 | 2.97 | 8.38 | 1.01 | 9.75 | 9.96 |
| | 2004 | 9.45 | 11.64 | 1.36 | 2.28 | 0.53 | 29.69 | 10.54 |
| | 2007 | 12.97 | 15.15 | 1.76 | 1.30 | 0.55 | 33.57 | 10.75 |
| | 2010 | 14.25 | 15.63 | 1.74 | 0.62 | 0.41 | 33.54 | 8.33 |
| | 2013 | 15.48 | 20.06 | 1.53 | 0.60 | 0.30 | 37.98 | 7.60 |
| | 2016 | 28.47 | 27.03 | 2.10 | 0.26 | 0.44 | 19.66 | 7.49 |
| **Total P Input** | | | | | | | | |
| | 1995 | 5112.55 | 1812.51 | 509.04 | 4169.55 | 201.40 | 2190.59 | 774.14 |
| | 1998 | 5313.08 | 2031.73 | 690.45 | 4328.45 | 148.78 | 1561.16 | 1066.57 |
| | 2001 | 5181.63 | 2828.00 | 697.92 | 2661.61 | 236.28 | 3511.46 | 2109.19 |
| | 2004 | 2732.58 | 2844.33 | 340.17 | 773.49 | 134.28 | 11418.04 | 2432.56 |
| | 2007 | 3393.35 | 3342.80 | 402.33 | 398.56 | 124.59 | 11741.41 | 2183.62 |
| | 2010 | 3444.85 | 3036.75 | 320.50 | 157.94 | 75.14 | 10516.22 | 1472.25 |
| | 2013 | 3504.81 | 3215.88 | 266.62 | 140.70 | 53.24 | 10639.94 | 1142.16 |
| | 2016 | 6176.63 | 4243.51 | 367.46 | 65.72 | 75.74 | 5758.85 | 1110.02 |
| **P Output (t)** | | | | | | | | |
| | grain | | | | | | | |
| | 1995 | 2155.15 | 1007.60 | 223.43 | 660.38 | 38.97 | 153.00 | 336.89 |
| | 1998 | 2550.20 | 1093.80 | 343.94 | 729.37 | 44.96 | 125.29 | 697.89 |
| | 2001 | 1981.60 | 1217.96 | 264.95 | 303.30 | 53.18 | 357.21 | 1023.33 |
| | 2004 | 936.73 | 1046.34 | 104.40 | 84.32 | 28.91 | 836.97 | 1024.55 |
| | 2007 | 1419.91 | 1489.62 | 167.79 | 59.77 | 31.16 | 902.60 | 1119.61 |
| | 2010 | 1647.00 | 1642.40 | 177.31 | 29.57 | 23.60 | 921.18 | 953.65 |
| | 2013 | 1705.00 | 1388.83 | 105.40 | 17.93 | 13.28 | 744.31 | 794.62 |
| | 2016 | 2539.00 | 1838.00 | 134.74 | 8.01 | 20.58 | 559.26 | 604.62 |

| P Input (t) | | Wheat | Maize | Rice | Soybean | Peanut | Cotton | Fruit-Vegetable |
|---|---|---|---|---|---|---|---|---|
| **Table 2 (continued)** | | | | | | | | |
| | Discarded straw | | | | | | | |
| | 1995 | 189.65 | 243.71 | 40.04 | 194.15 | 9.35 | 44.13 | 0.00 |
| | 1998 | 224.42 | 264.56 | 55.89 | 214.44 | 10.79 | 36.14 | 0.00 |
| | 2001 | 52.31 | 88.38 | 12.92 | 46.71 | 3.83 | 30.91 | 0.00 |
| | 2004 | 24.73 | 75.93 | 5.09 | 12.98 | 2.08 | 72.43 | 0.00 |
| | 2007 | 37.49 | 108.09 | 8.18 | 9.20 | 2.24 | 78.11 | 0.00 |
| | 2010 | 43.48 | 119.18 | 8.64 | 4.55 | 1.70 | 79.72 | 0.00 |
| | 2013 | 45.01 | 100.78 | 5.14 | 2.76 | 0.96 | 64.41 | 0.00 |
| | 2016 | 67.03 | 133.37 | 6.57 | 1.23 | 1.48 | 48.40 | 0.00 |
| | Erosion and runoff | | | | | | | |
| | 1995 | 129.40 | 34.92 | 22.02 | 86.83 | 5.76 | 42.80 | 23.35 |
| | 1998 | 153.47 | 68.78 | 22.58 | 100.89 | 4.83 | 32.76 | 37.83 |
| | 2001 | 123.01 | 76.07 | 18.59 | 52.51 | 6.30 | 61.13 | 62.40 |
| | 2004 | 57.22 | 70.47 | 8.21 | 13.82 | 3.20 | 179.69 | 63.82 |
| | 2007 | 90.73 | 105.96 | 12.34 | 9.06 | 3.83 | 234.81 | 75.18 |
| | 2010 | 116.45 | 127.72 | 8.45 | 3.90 | 2.14 | 274.12 | 68.03 |
| | 2013 | 142.58 | 154.24 | 11.76 | 4.62 | 2.33 | 292.05 | 49.25 |
| | 2016 | 202.72 | 192.45 | 16.81 | 2.56 | 2.66 | 139.99 | 40.11 |
| **Total P Output** | | | | | | | | |
| | 1995 | 2474.20 | 1286.23 | 285.48 | 941.36 | 54.08 | 239.93 | 360.24 |
| | 1998 | 2928.08 | 1427.14 | 422.41 | 1044.70 | 60.58 | 194.20 | 735.72 |
| | 2001 | 2156.93 | 1382.41 | 296.46 | 402.52 | 63.31 | 449.25 | 1085.73 |
| | 2004 | 1018.68 | 1192.74 | 117.70 | 111.12 | 34.19 | 1089.09 | 1088.37 |
| | 2007 | 1548.12 | 1703.66 | 188.32 | 78.03 | 37.23 | 1215.52 | 1194.79 |
| | 2010 | 1806.93 | 1889.29 | 194.40 | 38.03 | 27.44 | 1275.02 | 1021.68 |
| | 2013 | 1892.59 | 1643.84 | 122.30 | 25.32 | 16.57 | 1100.77 | 843.86 |
| | 2016 | 2808.75 | 2163.82 | 158.12 | 11.81 | 24.72 | 747.65 | 644.73 |

## PUE

Next, we examined the PUE of the studied crops. The PUEs of fruits and vegetables, and wheat and maize were higher than that of the other crops (Fig. 4C), while the PUE of cotton was the lowest (~6.9–10.2%). From 1998 to 2004, the PUEs of studied crops showed a slow downward trend 36.9% to 19.6% (Fig. 4C). However, after 2004, it steadily increased to 24.1%, 28.4%, 25.2%, and 32.1% in 2007, 2010, 2013, and 2016, respectively (Fig. 7).

Here, we performed PLS-PM structure equation models to explore the effects of P inputs and outputs on PUE and P load (Fig. 8). In the final PLS-PM model after removing the variables with loading values <0.7, the fertilizer block included the factors of residents' excrement, livestock excrement, and chemical fertilizers. The harvested grains was the most important factor affecting the PUE of crops (path coefficient = 0.934, $P < 0.05$), followed by fertilizers (path coefficient = −0.597, $P < 0.05$). Moreover, the fertilizers

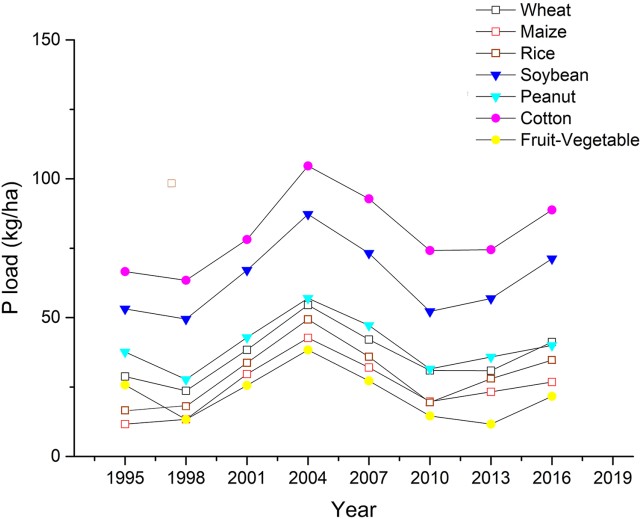

**Figure 6 Phosphorus (P) load of the seven crops in Dongying district from 1995 to 2016.** The crops include wheat, maize, rice, soybean, peanut, cotton and fruit-vegetable.

**Table 3 The P loading rates per unit area from the crop s and environmental risk assessment.**

| Year | Wheat | | Maize | | Rice | | Soybean | | Peanut | | Cotton | | Fruit-Vegetable | |
|------|-------|---------|-------|---------|------|---------|---------|---------|--------|---------|--------|---------|-----------------|---------|
| | R | Classes | R | Classes | R | Classes | R | Classes | R | Classes | R | Classes | R | Classes |
| 1995 | 0.53 | II | 0.33 | I | 0.51 | II | 0.63 | II | 0.50 | II | 0.68 | II | 0.50 | II |
| 1998 | 0.45 | II | 0.38 | I | 0.39 | I | 0.55 | II | 0.40 | I | 0.60 | II | 0.40 | I |
| 2001 | 0.57 | II | 0.48 | II | 0.51 | II | 0.68 | II | 0.51 | II | 0.74 | III | 0.51 | II |
| 2004 | 0.76 | III | 0.63 | II | 0.67 | II | 0.88 | III | 0.67 | II | 0.97 | III | 0.67 | II |
| 2007 | 0.65 | II | 0.52 | II | 0.56 | II | 0.76 | III | 0.56 | II | 0.84 | III | 0.56 | II |
| 2010 | 0.51 | II | 0.38 | I | 0.35 | I | 0.53 | II | 0.36 | I | 0.66 | II | 0.42 | II |
| 2013 | 0.53 | II | 0.35 | I | 0.40 | I | 0.55 | II | 0.40 | I | 0.64 | II | 0.38 | I |
| 2016 | 0.56 | II | 0.38 | I | 0.46 | II | 0.68 | II | 0.42 | II | 0.75 | III | 0.41 | II |

Notes:
Here, R repensents P load risk index;
$R \leq 0.4$, I Class, pollution-free;
$0.4 < R \leq 0.7$, II Class, slight pollution;
$0.7 < R \leq 1.0$, III Class, general pollution;
$1.0 < R \leq 1.5$, IV Class, serious pollution.

positively affected the P load, harvested grains, and P erosion and runoff ($P < 0.05$). Overall, fertilizer block is the key element in the PLS-PM structure equation model as it not only affects the PUE and P load but also affects the factors such as harvested grains and P erosion and runoff. Therefore, to improve the PUE, the comparatively effective method is to optimize fertilizer usage in agricultural soils whilst maintaining crop yields.

## DISCUSSION

### The application of PLS-PM in substance flow analysis

Phosphorus is not only a crop growth-limiting nutrient in estuaries but also an essential element for eutrophication in estuarine delta ecosystems (*Koh, 2019*; *Wu et al., 2019*).
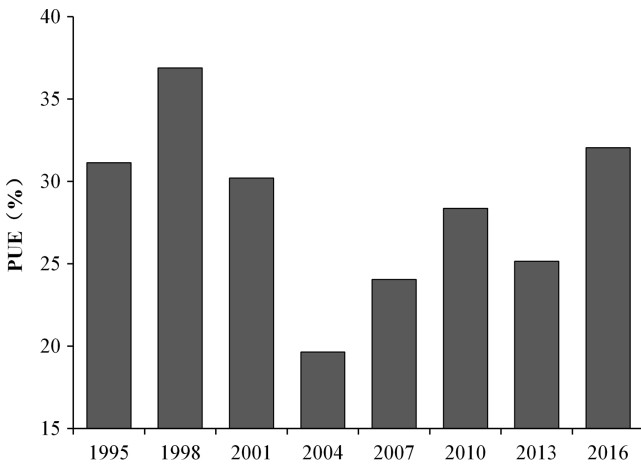

**Figure 7** Temporal trend of the phosphorus use efficiency (PUE) of the different crops in Dongying district from 1995 to 2016.                                   

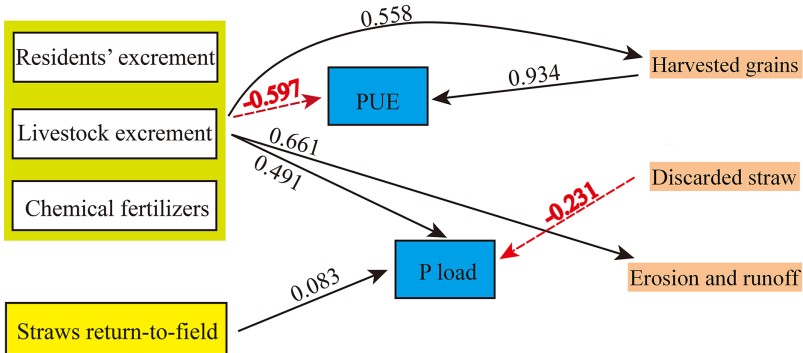

**Figure 8** The final partial least squares path models (PLS-PM) showing direct and total effects of significant factors of Phosphorus (P) input and Phosphorus (P) output on the phosphorus use efficiency (PUE) and Phosphorus (P) load in crops. These significant factors were divided into five block variables: fertilizers (residents' excrement, livestock excrement and chemical fertilizers), straws return-to-field, harvested grains, discarded straw and Phosphorus (P) erosion and runoff. Here, only paths with the significance $P < 0.05$ are shown for simplicity. Black solid and red dashed lines represent positive and negative effects, respectively. The absolute value of the path coefficients are shown on the lines.                                   

The P enrichment in sediments from eroded upstream water and upland sources can significantly affect the crop yields and eutrophication risks of coastal areas (*Qu et al., 2021*).

Notably, the Yellow River carries $1.08 \times 10^9$ t of fluvial sediment annually to the Bohai Sea (*Milliman & Meade, 1983*). The sediment deposition has formed the Yellow River Delta which is recognized as the most active coastal zone with land-ocean interaction (*Qu et al., 2021*). The Dongying District, the core region of the Yellow River Delta, is located at the intersection of the Shandong Peninsula and the Bohai Sea and has been developed into a major agricultural production base (*He et al., 2020a*). In order to increase crop yield, P fertilizers were excessively used which increased the P accumulation in soils, seawater, and ultimately led to eutrophication of the Bohai sea (*Xu et al., 2020*).
The typical soil type in the Dongying District is classified as saline alluvial soil (*Meng et al., 2020*). Notably, excessive application of P fertilizers aggravated soil salinization and ultimately reduced PUE in the Dongying District. The Dongying government firmly implements the guidelines of green agriculture development which encourage the farmers to reduce chemical fertilizers and improve PUE (*Xu et al., 2020*). To improve PUE, it is necessary to understand its causal relationship with P inputs and outputs. Here, we used the PLS-PM model to quantify the contributions of P inputs and outputs to PUE in the agricultural system of Dongying District.

PLS-PM is a variance-based structural equation modeling technique that relies on an alternating least squares algorithm (*Henseler, 2018*). It is widely used in various disciplines as an effective tool to analyze the relationships and influence of different aspects on an observed phenomenon. Recently, the PLS-PM method has been further improved for the analysis of cause-effect relationships in confirmatory and explanatory research (*Benitez et al., 2020*). In our study, the fertilizers, straws return-to-field, harvested grains, discarded straw, and P erosion and runoff were defined as block variables to estimate the relationship with PUE and P load using the improved PLS-PM method. The PLS-PM model was run based on the bootstrap procedure to validate the path coefficients and the coefficients of determination.

Our study is the first to use the PLS-PM model for phosphorus flow analysis. Moreover, we established the numerical relationship between the abovementioned five block variables, PUE, and P load, which is the most significant improvement over other studies. In agreement with previous studies (*Wu et al., 2016*), we found that fertilizer and crop production are the key factors affecting the PUE of the Yellow River Delta region. We suggest that optimizing fertilizer application rate and increasing harvest can effectively increase the PUE. This is a feasible approach for soil P management. We summarize some recommendations with discussion in "Data uncertainties".

## Comparison of PUE between Yellow River Delta and other deltas

The PUE of crops in the Yellow River Delta was compared with the reported PUEs of other river deltas (Table 4). The crop species, research methods, boundaries and research periods in those studies are listed in Table 4. We found that the PUEs of crops in the Yangtze and Pearl River Deltas were generally lower than that of the Yellow River Delta. This phenomenon can be attributed to P sorption and P forms in soils. Soil inorganic P could account for up to 50–90% of soil total P (*Feng et al., 2016*). The soil inorganic P includes the compounds Al-P and Fe-P in acidic soils and the compounds Ca-P in calcareous soils (*Meena et al., 2018*). According to the P transforming dynamics, the soil inorganic P could be fractionated into the available P (Olsen P), moderate-cycling P (variscite (Al-P), strengite (Fe-P), dicalcium phosphate (Ca2-P) and octacalcium phosphate (Ca8-P)), and recalcitrant P fractions (hydroxyapatite (Ca10-P) and occluded P (O-P)). Olsen P, which can be directly absorbed by crops, is closely related to PUE. The Olsen-P content showed great differences among deltas (*Li et al., 2015*). It was higher in the Yellow River Delta than in the Yangtze and Pearl rivers deltas. Specifically, the Olsen-P content in >90% of agricultural soils in Pearl River Delta is lower than the
**Table 4 Comparison between PUE of different agricultural system.**

|  | This study | *Zheng et al. (2017)* | *Ma et al. (2012)* |
|---|---|---|---|
| Crops | Wheat, Maize, Rice, Soybean, Peanut, Cotton, Fruit-Vegetable | Rice, wheat, maize | Crop |
| Methods | SFA method | SFA method; Multi-modal neural networks | NUFER model |
| Region | Dongying district, Shandong Province, Yellow River Delta | Around Yangtze River Delta, China | In Guangdong Province, Pearl River Delta |
| Boundary | Crop systems | Crop systems | Crop systems |
| Year/period | 1995–2016 | 2001–2015 | 2005 |
| PUE (%) | Wheat (42.76%), Maize (46.80%), Rice (42.51%), Soybean (14.21%), Peanut (25.26%), Cotton (8.21%), Fruit-Vegetable (54.96%) | 53.61% (rice), 36.22% (wheat), 32.56% (maize) | 27% |

**Note:**
SFA, substance flow analysis; NUFER, Nutrient flows in food chains, environment and resources use.

recommended value (39 mg kg$^{-1}$) for crop production (*Li et al., 2015*). In Pearl and Yangtze rivers deltas, the P-deficient land accounts for >50%, compared with 46% in the Yellow River Delta. That explains the reason for lower PUE in Yangtze and Pearl rivers deltas than in the Yellow River Delta.

The distribution and morphology of soil P are the key factors affecting PUE. Although the mineral forms of phosphate are not readily available to crops, Al-P, Fe-P, Ca2-P and Ca8-P can be transformed into free P forms as important buffering pools for Olsen P. Al-P and Fe-P are the main P forms in acidic soil of Pearl River Delta, while Ca2-P and Ca8-P are prominent in calcareous soils of Yellow River Delta (*Xu et al., 2019*). Notably, dissolution of Ca-P compounds is the main approach for P release under acidic conditions, which is related to ion exchange between OH- and the Fe-P and Al-P compounds (*Huang et al., 2021*). Furthermore, the Fe-P/Ca-P ratio significantly affects the PUE. Soils with higher Fe-P/Ca-P ratios release more P under alkaline conditions, and in turn higher PUE (*Huang et al., 2005*). The soil system is a complex dynamic ecosystem (*Chen et al., 2002*). Thus, to improve PUE, a better understanding of the distribution and morphology of P is important to frame scientific and sensible guidelines.

## Data uncertainties

The data and parameters used in the SFA model on P flows were mainly obtained from the statistical yearbooks of the Dongying District, face-to-face interviews, questionnaires, and research articles. The data from the official yearbooks are publicly available and are frequently used in the research of material flow analysis (*Han et al., 2021*). The local statistician may have changed the statistical methods during the period of 22 years. Consequently, the uncertainty level is greater for the data from the older yearbooks.

The parameters such as the P-containing rates of crop seeds ($r^{seed}$), grain to straw ratio of crops ($r^{grain-straw}$), P-containing rate of crop straws ($r^{straw}$), P content of residents' excrement ($w^{resid}$), P content of livestock excrement ($w^{livestock}$), P-containing rate of grains

($r^{grain}$), were obtained from previous research articles (Table S1), with the limitation of space and time, which could inevitably introduce errors into our study (*Wu et al., 2016*).

Moreover, the parameters including average wind erosion intensity per sown area ($w^{wind}$), P content in the rainfall ($r^{rain}$), P loss content per unit of crop area ($r^{loss}$) are mainly based on the local natural environment and climate. These data are difficult to get and relatively uncertain. Furthermore, the data obtained from the interview and questionnaire, for example, amounts of crop seeds per sown area ($w^{seed}$), amount of compound fertilizer used ($B^{com}$), amount of phosphate fertilizer used ($B^{phos}$), straw return-to-field ratio ($r^{retured\ straw}$), proportion of residents' excrement applied to the field ($r^{resid}$), and proportion of livestock excrement applied to the field ($r^{livestock}$) might only have reflected the farmland situation of interviewee rather than the whole field information of Dongying District. Although we consider these uncertainties because of data limitations, the uncertainties could not be fully eliminated but were largely minimized by comparing the related studies and broadening the investigation. Additionally, the local monitoring and field experimentation could have improved the parameters and data accuracy (*Wu et al., 2016*).

## Suggestions for improving PUE

We found that the PUEs of studied crops were low (~20–40%) and differed among different crops. The PUE of cotton was the lowest (~6.9–10.2%). Improving the PUE of cotton can improve P management in the agricultural systems of the Yellow River Delta. Additionally, the relationship between P input from fertilization and PUE was studied (Fig. S1). The PUE of wheat, rice and soybean positively correlated with P contents of fertilizers, the Pearson correlation coefficients ($r$) were 0.047, 0.16, and 0.25 for wheat, rice, and soybean respectively. However, the PUE of maize, peanut, cotton, and fruit-vegetables was negatively related with the P content of fertilizers ($r = -0.62, -0.58, -0.29, -0.44$ for maize, peanut, cotton, fruit-vegetables, respectively). The farmers of maize, peanut, cotton and fruit-vegetable should make the best use of fertilizers. Furthermore, our research suggests that the low PUE in the Yellow River Delta region is attributed to heavy application of fertilizers and low harvest. Here, we suggest the following measures to improve PUE.

### *Optimizing fertilizer application*

(1) Soil testing and fertilizer recommendation

Soil testing can determine the soil's fertility status to regulate fertilizer demand and improve fertilizer efficiency (*Liu et al., 2017*; *Sun et al., 2013*). Soil testing and fertilizer recommendation were implemented in the Dongying District from 2004. From then on, fertilizer inputs reduced by 25.83, 83.33, and 19.66 kg/ha for wheat, maize, and cotton fields in 2013, respectively. However, the soil testing and fertilizer recommendation can be significantly affected by several factors, such as local soil condition, climate situation, and cotton varieties (*Jordan-Meille et al., 2012*). Moreover, the structure of fertilizer can negatively impact the effect of soil testing and fertilizer recommendation. For example, the local farmers chose chemical fertilizers over organic fertilizers to save manpower.

The excessive use of chemical fertilizers decreases soil fertility and in turn PUE. Therefore, we suggest that the soil properties (pH, clay content), climate situation, cotton species, and fertilizer structure should be considered to attain the optimum results (*Jordan-Meille et al., 2012*).

(2) Modifying the P-fertilizer treatment

In the Dongying District, P is applied by the broadcast fertilization method. However, the method is not effective for P uptake by crops (*Schroder et al., 2011*). Thus, to increase PUE, novel scientific fertilization modes should be encouraged in the Dongying District. A study showed that lowland rice soil of Sub-Saharan Africa needed twice the amount of P fertilizer if broadcasted instead of P-dipping (*Rakotoarisoa, Tsujimoto & Oo, 2020*). Here, we recommend the root-zone P fertilization instead of broadcast fertilization to improve P fertilizer efficiency in the Dongying District.

(3) Combination of inorganic fertilizer and organic fertilizer

In the Dongying District, the amounts of organic fertilizers (for example, excrement) reduced gradually, while the amount of chemical fertilizer increased rapidly from 1995 to 2016 (Fig. 4A). The organic fertilizers, such as excrement, increase carbon and organic matter in soil that can improve soil P binding, retention, and availability to crops (*Yang, Chen & Yang, 2019*). However, long-term application of organic fertilizer would promote the soil accumulation of heavy metals, *e.g.*, Zn, Cd, and Cr, inducing adverse effects on food safety (*Ning et al., 2017*). Thus, we suggest that the farmers of the Dongying District should have the reasonable application of organic fertilizers.

(4) Optimizing the planting structure

We found that the cotton and soybean had low PUE and high P load. To improve PUE, the growing area of the cotton and soybean should be reduced appropriately, whilst the growing area of the crops with high PUE and low P load should be increased. However, the local agricultural structure is mainly dependent on the market economy and local planting habits, and therefore cannot be fully controlled.

### Increasing the harvest

(1) Utilizing P solubilizing microorganisms (PSMs)

Many studies have reported the solubilization of insoluble P by PSM (*Richardson, 2001*; *Sharma et al., 2013*), which can convert unavailable P into available P (Olsen P) improving P uptake by crop and higher yield. Generally, PSMs can be divided into two classes; (1) inorganic P-solubilizing microorganisms secreting organic acid to dissolve inorganic P compounds, and (2) organic P-mineralizing microorganisms secreting phosphatase to enzymatically mineralize organic P compounds (*Alori, Glick & Babalola, 2017*). Both field and pot experiments with or without P-fertilizers showed that PSMs can improve crop yield and P uptake (*Bolo et al., 2021*; *Jiang et al., 2021*). For example, *Noor et al. (2017)* used the co-application of P fertilizer-PSM (*Pseudomonas putida*) to improve corn dry matter yield and P uptake by 12% and 33%, respectively.

Additionally, PSMs decrease soil pH and are better suitable for natural and alkaline soils. Therefore, PSMs can greatly improve available P and PUE without influencing the soil health in the Yellow River Delta region. Various soil species of PSMs (including both

fungi and bacteria) can solubilize phosphorus. However, it is difficult to predict effective PSMs in each field. Therefore, screening effective PSMs is necessary for a specific location.

(2) Improving the mechanization level

The crop harvest is significantly affected by the mechanization level. Specifically, in cotton planting, the increase in labor cost and lower production restricts productivity which can be improved with mechanization. The farmers in the Shawan county of Xinjiang Province have realized the cotton mechanization from sowing to harvesting. The local government and farmers introduced a new packing cotton picker, and the process of cotton harvesting and packaging has become highly mechanized. This greatly improved the cotton yield and quality. We suggest that the farmers in the Dongying District should establish similar cotton production mechanization, and learn from the cotton planting experience of Xinjiang Province.

The abovementioned strategies can provide the possibilities to reduce P load and improve PUE, however, with farmer's requirements and acceptance. The ecological agriculture development in the Dongying District is largely dependent on the collaborative partnerships of policymakers, researchers, and farmers. Additionally, the factors of government policy, price, cropping mechanization, and farmer's motivation would indirectly affect PUE by influencing the cultivation area. For example, the government policy of encouraging local farmers to extend cotton plantation since 2004, the supply of maize exceeds demand in Dongying District. However, only a few farmers were willing to plant soybean, because of tedious harvesting, storage and post-harvest management, as well as low economic efficiency (*Bern, Hanna & Wilcke, 2008*). Furthermore, the low prices and cropping mechanization muted farmers' enthusiasm to grow wheat and corn from the year of 1995 to 2004, reducing the total P output of these crops. Therefore, the researchers should fully consider those factors, *e.g.* government policy, price, cropping mechanization, and farmer's motivation, in further study of how to improve PUE.

## CONCLUSIONS

In this paper, we used SFA to establish the dynamic model of phosphorus flow in different crops of the Dongying District from 1995 to 2016. We found that P input increased steadily from 1995 to 2007, and then decreased from 2010 to 2016. Specifically, compared with other crops, cotton had the highest P input from 2004 to 2013. Among the contributing sources of P input, chemical fertilizers contributed the highest. Meanwhile, the P amount from harvested grains accounted for 80–90% of the total P output.

Also, the cotton had the highest P load and topped in the P load risk ranking. Excessive P load in cotton caused problems with the plant uptake of elements and potentially increased eutrophication risk. Additionally, the PUE was significantly different among distinct crops. The PUE of cotton was the lowest. Based on the PLS-PM structure equation model, fertilizers and harvested grains were the two important factors affecting PUE. Therefore, reducing the use of fertilizer in agricultural soils whilst maintaining crop yields can effectively improve PUE.

Finally, we provide some recommendations for improving PUE and reducing P load in agricultural soil. However, the soil physicochemical properties and biases of cotton

planting in the Dongying District should be considered before implementing recommendations. Additionally, farmers' requirements and acceptance should be respected. In-field demonstrations and formulating the fertilization scheme can significantly improve policy implementation.

### Funding
This study was supported by the National Key Research and Development Program of China (2018YFD0800303), the National Natural Science Foundation of China (41977144 and 31970545), and the Postdoctoral Science Foundation of Qingdao City (61460079311133). The funders had no role in study design, data collection and analysis, decision to publish, or preparation of the manuscript.

### Grant Disclosures
The following grant information was disclosed by the authors:
National Key Research and Development Program of China: 2018YFD0800303.
National Natural Science Foundation of China: 41977144, 31970545.
Postdoctoral Science Foundation of Qingdao City: 61460079311133.

### Competing Interests
The authors declare that they have no competing interests.

### Author Contributions
- Huan He conceived and designed the experiments, performed the experiments, analyzed the data, prepared figures and/or tables, authored or reviewed drafts of the paper, and approved the final draft.
- Lvqing Zhang conceived and designed the experiments, performed the experiments, analyzed the data, prepared figures and/or tables, and approved the final draft.
- Hongwei Zang conceived and designed the experiments, analyzed the data, authored or reviewed drafts of the paper, and approved the final draft.
- Mingxing Sun analyzed the data, prepared figures and/or tables, and approved the final draft.
- Cheng Lv performed the experiments, prepared figures and/or tables, and approved the final draft.
- Shuangshuang Li performed the experiments, authored or reviewed drafts of the paper, and approved the final draft.
- Liyong Bai analyzed the data, prepared figures and/or tables, and approved the final draft.
- Wenyuan Han performed the experiments, analyzed the data, prepared figures and/or tables, authored or reviewed drafts of the paper, and approved the final draft.
- Jiulan Dai conceived and designed the experiments, analyzed the data, authored or reviewed drafts of the paper, and approved the final draft.

## Data Availability

The raw measurements are available in Table 2 and the Supplemental Files.

## Supplemental Information

Supplemental information for this article can be found online at http://dx.doi.org/10.7717/peerj.13274#supplemental-information.

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
