# Peer review of "Phosphorus flow analysis of different crops in Dongying District, Shandong Province, China, 1995–2016"

_PeerJ, doi:10.7717/peerj.13274_

## Round 0.1 · original submission · Major Revisions

Reviewers' comments on your work have now been received. The manuscript has been assessed by three reviewers. Reviewers indicated that the experimental design and statistical analysis should be improved. Moreover, the available raw data should be provided. I agree with this evaluation and I would, therefore, request for the manuscript to be revised accordingly.

Reviewer 1 ·

Basic reporting

This manuscript is clearly written in professional, unambiguous language. The author focused on long-term P flow in Dongying district, a core region of the alluvial delta at the estuary of the Yellow River. It is interesting to know the status of P accumulation and PUE in a long period (22 years) at such a typical alluvial delta. However, this study mainly depended on data collected from different sources, the author should provide all the available raw data used for the analyses. There are too many separated figures, some of them (e.g. Fig. 4,5, and 8) can be shown together in one figure. Fig. 4 and Table 3 were not well labelled and described (Please see General comments),

Experimental design

This research is about environmental ecology which is within the scope of the PeerJ, and the research question about “long-term P flow of different cropping systems at alluvial delta” is also important to assess the impact of P fertilization on environment. However, some of the data is from interviews, questionnaire, and published data, the author should give details about how to get the data.

Validity of the findings

It is a common sense that fertilizer and crop yield are key factors affecting PUE, the PLS-PM model doesn’t provide more new findings. Because the author focused on the PUE of different cropping systems, it is interesting to know the relationship between P fertilization rate and PUE in different cropping systems over the 22 years, from which some practical guidance might be made to local farmers. The author should also discuss the limits of this study due to data uncertainties.

Additional comments

Line 102 reference Yuan et al. (2011) , please delete Line 106 (Yuan et al., 2011)
Line 108 reference same problem as shown in Line 102
Line 117 What does the author mean ‘gain crops’? Should it be ‘grain crops’?
Line 119 doesn’t make sense “ Why the soil characteristics and crop-planting structure forced excess P fertilizer usage?”
Line 135 – 139 This information is necessary, it should be removed.
Line 141 planning? Did you mean ‘planting’?
Line 143 gain? ‘grain’
Line 150 Please indicate which four cropping systems.
Line 151-152 The English language should be improved to make it more clear.
Line 167 What do you mean ‘the risk model of P load’? More details are required to explain.
Line 180 ‘those’ should be ‘these’
Line 190 More details should be provided about how you collected data.
Line 196 The data indicated by author is not listed in table 1
Line 211-212 The author should explain how the straw return-to-field ratios were calculated? The ratios should be related to the area the author focused on, not from the reference.
Line 337-338 This is a well-known conclusion about how to increase PUE. The author should edit this part.
Line 377-378 How does the author get this conclusion? The application of fertilizer and crop yield should have positive linear correlation at certain range, but the PUE might decrease along with the increased fertilizer.
Line 389 ‘fertilization’ should be ‘fertilizer’?
Line 413 ‘planning’ should be ‘planting’?
Line 432 need reference
Line 420-437 PSMs are not always effective, especially in field. You should mention this point. Screening effective PSMs is necessary for specific location.
Line 457-459 The P input is related to cropland size, so total P application rate should be more meaningful regarding to PUE.

Fig. 4 Why only three input factors were shown in the figure, but 7 factors were shown in the legend.
Too many separated figures, some of them (e.g. Fig. 4,5, and 8) can be shown together in one figure
Table 3 Classes I, II, and III should be indicated respectively.

Reviewer 2 ·

Basic reporting

The article requires major revisions before it can be published in PeerJ. The authors need to clearly indicate the statistical design that was adopted in the study. There is also no proof of model validation which is very important to determine the accuracy of the current results. There are several minor errors throughout the manuscript that need to be corrected as indicated in the additional comments section. The results section also needs to be revised to distinguish it from the discussion section.

Experimental design

The experimental design is not quite clear and the authors need to clarify which design was adopted in the study. The authors also referred to different crops e.g. cotton, corn, etc as cropping systems. This should be corrected to crops and not cropping systems. It is important that the authors clarify the cropping systems (organic or convention) each crop was grown under and this will help point the factors that contributed to the differences observed in different crops with regards to PUE.

Validity of the findings

The authors indicated that this study was the first to use the PLS-PM to quantify the contributions of P inputs and outputs to PUE in the agricultural system of Dongying District. However, there is no proof of model validation to check the accuracy and performance of the model basis on the past data for which there are already actuals. This will also determine the accuracy of the results of the current study.

Additional comments

There are several other minor but several corrections required throughout the manuscript as listed below.
Line 26: Define P for the first time.
Line 28: Cotton is a crop and not a cropping system.
Line 29: Define PUE for the first time.
Line 64: What is the PLS-PM model?
Line 74: Remove the word flow after P.
Line 84: Remove the word respectively appearing before (Jiang et al...)
Line 88-98: Write PUE in full when appearing at the beginning of the sentence and it should be P fertilizer and not fertilizer P.
Line 91: research and not researchers.
Line 102: Write SFA in full when appearing at the beginning of the sentence.
Line 108: Li et al. (year) and remove the citation at the end of the sentence.
Line 116: Yields for which crop?
Line 146: Please insert the word ''year'' before 2000.
The results section reads like the discussion section. Please discuss your results in the discussion section and not in the results section.
Line 225 and 305: Write P in full when appearing at the beginning of a subsection or sentence.

·

Basic reporting

no comment

Experimental design

no comment, does not apply

Validity of the findings

The term “P Use Efficiency (PUE)” is used in literature for different levels and different methods of calculation as a way to calculate how efficient the P use is in a system. There is some unclearness about the calculation of the PUE in the study.
What I understand from the paper in Line 87-88:
In the manuscript it is suggested that in the paper the PUE that is used is defined as increase in crop yield by the application of P. For me this suggests that the yield without P is subtracted from the yield with P and divided by the P fertilization. This is a legitimate way to calculate the PUE and is often indicated as Apparent P Recovery (APR).
This is, however, not calculated in this paper. In this paper the PUE is calculated by input/output (in which output includes P losses by erosion and runoff). This is also a legitimate way of calculating the PUE but this is not what is described.
In this FAO bulletin the different methods of PUE calculation are described:
https://soil5813.okstate.edu/Spring2012/Syers%202008.pdf

Quote: “There are a number of agronomic indices and methods for measuring the efficiency of plant-nutrient use in agriculture. In summary, the methods and indices, based on those of Cassman et al. (1998), are: direct method; difference method; partial factor productivity index; physiological efficiency index; and balance method.”
“.... a balance method is used here, i.e. total P in the crop divided by the P applied (UP/FP)”

This is closer to what is calculated in this study.
However, in this study it looks like the losses are also considered as the output that is used to calculate PUE, line 157-158. If that is so: In my opinion this is not correct to use in defining the PUE: an efficiency would be the ratio of USEFUL output/input. If losses are included in the output the PUE would go up if the losses are higher. This is in my opinion not what you want to express in an efficiency indicator.
If losses are not used for output in the calculations of PUE: please mention that in Materials and Methods.

Additional comments

Line 103 and 107: please add year after references.
Line 143 : gain should be grain (I assume)
Line 156 : straw return to field: it is not clear from figure 2 what is the origin of this flow. Is it an internal flow? Harvested from the system and brought back into the system? The it should be also taken up as output. In the text in line 157 it seems to be correct, there all harvested straw is mentioned as output. Please make text and figure 2 correspond.
Line 163: dynamic : in what sense is it dynamic: in time?
Line 204: 0.45: what unit?
Line 206/207 and 208/209 : rates: what units? %?
Line 235: decrease of soybean is caused by decreasing area or yield per area? I presume area, please mention that.
Line 239: unbalanced: in what aspect?
Line 164: planting area: would “cropping area” be a better term? (not sure)
Line 323 – 331 move to Materials&Methods
Line 352/353: soil PUE: please define, what is the difference with crop PUE (line 342)
Line 391: how does excessive use of chemical fertilizers decrease soil fertility?
Line 398: P absorption: should be P uptake (I think)
Line 401: root P fertilization: what do you mean: fertilizer placement close to the roots?
Line 411: is more organic fertilizer available? How about other nutrients in organic fertilizer? Should there be warned for excess of other nutrients and thus decrease the use efficiency of other nutrients?
Line 430: dry matter: add “yield”

---

## Round 0.2 · Minor Revisions

Reviewers' comments on your work have now been received. The manuscript has been assessed by three reviewers. Reviewers indicated that there was still some language issues in the current manuscript. I agree with this evaluation and I would, therefore, request for the manuscript to be revised accordingly.

Reviewer 1 ·

Basic reporting

The authors have answered my questions and comments. I agree to publish this paper in Peerj, once the following minor issues are addressed.

91: police makers, policymakers
249: Of, In
313: P, Phosphorus (Abbreviation normally is not allowed at the beginning of a sentence)
348-349: Reducing fertilizer application rate might decrease harvest. It would be a better way to say “optimizing fertilizer application rate to maximize PUE”
352: betweenYellow between Yellow
406: What do the authors mean “the same”?
414: Supplementary figure 1 needs figure caption.

Experimental design

NA

Validity of the findings

NA

Additional comments

NA

·

Basic reporting

no comment after revision

Experimental design

no comment after revision

Validity of the findings

no comment after revision

---

## Round 0.3 · accepted · Accept

The author has modified the paper and is ready for publication.